# Chromium-Ruthenium Oxides Supported on Gamma-Alumina as an Alternative Catalyst for Partial Combustion of Methane

**Tanakit Chomboon [1], Weerit Kumsung [1], Metta Chareonpanich [1,2,3], Selim Senkan [4] and Anusorn Seubsai [1,2,3,*]** 

1   Department of Chemical Engineering, Faculty of Engineering, Kasetsart University, Bangkok 10900, Thailand; Tanakit.chom@ku.th (T.C.); kemweerit@gmail.com (W.K.); fengmtc@ku.ac.th (M.C.)
2   Center of Excellence on Petrochemical and Materials Technology and Research Network of NANOTEC–KU on NanoCatalysts and NanoMaterials for Sustainable Energy and Environment Kasetsart University, Bangkok 10900, Thailand
3   Research Network of NANOTEC–KU on NanoCatalysts and NanoMaterials for Sustainable Energy and Environment, Kasetsart University, Bangkok 10900, Thailand
4   Department of Chemical and Biomolecular Engineering, University of California Los Angeles, Los Angeles, CA 90095, USA; senkan@ucla.edu
*   Correspondence: fengasn@ku.ac.th

**Abstract:** Catalyst screening of $\gamma$-$Al_2O_3$-supported, single-metal and bimetallic catalysts revealed several bimetallic catalysts with activities for partial combustion of methane greater than a benchmark Pt/$\gamma$-$Al_2O_3$ catalyst. A cost analysis of those catalysts identified that the (2 wt%Cr + 3 wt% Ru)/$\gamma$-$Al_2O_3$ catalyst, denoted as 2Cr3Ru/$Al_2O_3$, was about 17.6 times cheaper than the benchmark catalyst and achieved a methane conversion of 10.50% or 1.6 times higher than the benchmark catalyst based on identical catalyst weights. In addition, various catalyst characterization techniques were performed to determine the physicochemical properties of the catalysts, revealing that the particle size of $RuO_2$ became smaller and the binding energy of Ru 3d also shifted toward a lower energy. Moreover, the operating conditions (reactor temperature and $O_2$/$CH_4$ ratio), stability, and reusability of the 2Cr3Ru/$Al_2O_3$ catalyst were investigated. The stability test of the catalyst over 24 h was very good, without any signs of coke deposition. The reusability of the catalyst for five cycles (6 h for each cycle) was noticeably excellent.

**Keywords:** catalyst; chromium; gamma alumina; partial combustion; methane; ruthenium

## 1. Introduction

Methane ($CH_4$) is a natural gas produced from the anaerobic decomposition of organic compounds. Due to the relative abundance of $CH_4$ on Earth, there is great interest in converting it into alternative fuels or energy sources. The partial combustion of $CH_4$ is an alternative to coal as a means of generating CO, $CO_2$ and $H_2$ [1]. CO and $H_2$ have many industrial applications. For example, CO can be used as a chemical feedstock for producing various hydrocarbons (e.g., aldehydes, phosgene, methanol). $H_2$ can be used for generating heat or electricity and generates zero carbon emissions, as well as being a very useful chemical in the chemical and petroleum industries such as in the production of polymers, light hydrocarbons, and ammonia.

The partial combustion of $CH_4$ occurs when the $O_2$/$CH_4$ ratio is less than the stoichiometric ratio [2–5]. The reaction without a catalyst can occur at high temperature (>700 °C). If a catalyst is used, the temperature is reduced and more methane is converted. Importantly, when the $O_2$/$CH_4$ ratio

is equal to or greater than the stoichiometric ratio, complete combustion (producing only $CO_2$ and $H_2O$) is favored [6]. In general, a solid catalyst that possesses high activity, stability, reusability, and more importantly has a low cost is desirable for $CH_4$ activation. In the past, several supported and unsupported solid catalysts have been extensively investigated. Pt [7,8] Rh [9,10] and Pd [11–14] are the best known transition metals to have been widely studied for the reaction. Among these three metals supported on $\gamma$-$Al_2O_3$ under fuel-rich conditions, the order of the catalytic activity for the metals has been ranked as: Pd > Rh > Pt [15]. Some bimetallic or multi-component catalysts have higher $CH_4$ conversion rates compared to their single-metal catalysts—for example, $Pd/Al_2O_3$ modified with NiO and MgO [16], Rh-$LaMnO_3$/$Al_2O_3$ [17], Ce-Cr/$\gamma$-$Al_2O_3$ [18], Ni-Cr/yttria-doped Ce [19], Rh-$LaMnO_3$ honeycomb [20], and Au-Pd/nanohybrid 3D macroporous $La_{0.6}Sr_{0.4}MnO_3$ [21,22]. In the presence of two or more main metal components, the reported activity of these catalysts was mostly enhanced due to a synergistic catalytic effect and/or the second metal added enhanced dispersion of the main active species [23].

Although various single-metal and bimetallic catalysts have been studied, systematic screening of such catalysts has not been fully explored and reported. In previous work, we adopted a combinatorial approach to explore new active bimetallic catalysts. We found several bimetallic catalysts that delivered a $CH_4$ conversion higher than that of the benchmark Pt catalyst [24]. These catalysts were discovered using a screening method in which 20 single noble metals supported on $\gamma$-$Al_2O_3$ were first screened to identify the six most-active metals at 475 °C and atmospheric pressure. Subsequently, the combinations of the six metals were prepared similarly and screened for the partial combustion of $CH_4$. The most active bimetallic catalyst discovered was the combination of Rh and Cr at a Rh/Cr/$Al_2O_3$ weight ratio of 4:1:95. Using this catalyst, $CH_4$ conversion was about 2.3 folds compared to the benchmark Pt catalyst. However, to date there has been no cost analysis of the prepared catalysts, which is a key factor to consider when using a catalyst in commercial applications. Consequently, in the current work, the estimated costs were carefully determined of active single and bimetallic catalysts. Of the more active bimetallic catalysts, the cheapest compared to the benchmark Pt catalyst was then chosen and, along with the single-metal catalysts of its components (Ru, Cr) and the benchmark Pt catalyst, it was characterized and investigated under various operating conditions to determine its potential as a commercial catalyst.

## 2. Results and Discussion

### 2.1. Activity and Cost of Single-Metal and Bimetallic Catalysts

In a previous report [24], 20 elements were selected for single-metal catalyst screening, and it was found that the six most active single catalysts were 5Rh > 5Ru > 5Pd > 5Pt > 5Cu > 5Cr, where the number indicates the weight percentage on the alumina support. The $CH_4$ conversions were in the range 4.64–9.34% and the testing conditions used were 475 °C, an $O_2$/$CH_4$ ratio of 3:5, and a total feed gas flow rate of 50 mL/min. Subsequently, bimetallic catalyst screening was performed by combining each of these six active metals with the others at five different weight ratios. It was found that the most-active bimetallic catalyst was the combination of 4 wt% Rh and 1 wt% Cr on the $Al_2O_3$ support. However, in addition to the activity for $CH_4$ conversion of those catalysts, cost was also a main focus in the present work. Therefore, the six most-active single-metal catalysts and all 60 bimetallic catalysts were plotted as a function of cost and $CH_4$ conversion, as shown in Figure 1. Note that the cost of the catalysts was calculated from the main active metal only, without adding other costs. Furthermore, the cost of each element was based on the average world metal prices during 2013–2018 [25–30]. The average prices of Rh [25], Pt [26], Pd [27], Ru [28], Cr [29], and Cu [30] were 40.290, 37.395, 28.007, 3.264, 0.393, and 0.006 USD/g of pure metal, respectively. Since Pt supported on $\gamma$-$Al_2O_3$ is widely used in the partial combustion of $CH_4$, 5 wt% Pt on $\gamma$-$Al_2O_3$ (5Pt/$Al_2O_3$) was used as a benchmark catalyst in this work. The benchmark catalyst gave a $CH_4$ conversion of 6.69% at an average cost of 1.867 USD/g of catalyst (hereafter denoted USD/g). The cost of the 4Rh1Cr/$Al_2O_3$

catalyst, which achieved the highest $CH_4$ conversion (15.77%), was 1.616 USD/g. Many other bimetallic catalysts supported on $Al_2O_3$, such as 3Pd2Rh, 2Pt3Rh, 1Cr4Pd, and 1Ru4Rh, had $CH_4$ conversion rates higher than that of the benchmark catalyst. However, their prices were almost identical to that of the benchmark catalyst. Accordingly, these catalysts were considered to be uneconomical for industrial use. It is important to emphasize that these comparisons were made in terms of identical total weights of the active elements in the catalysts and based on element prices only.

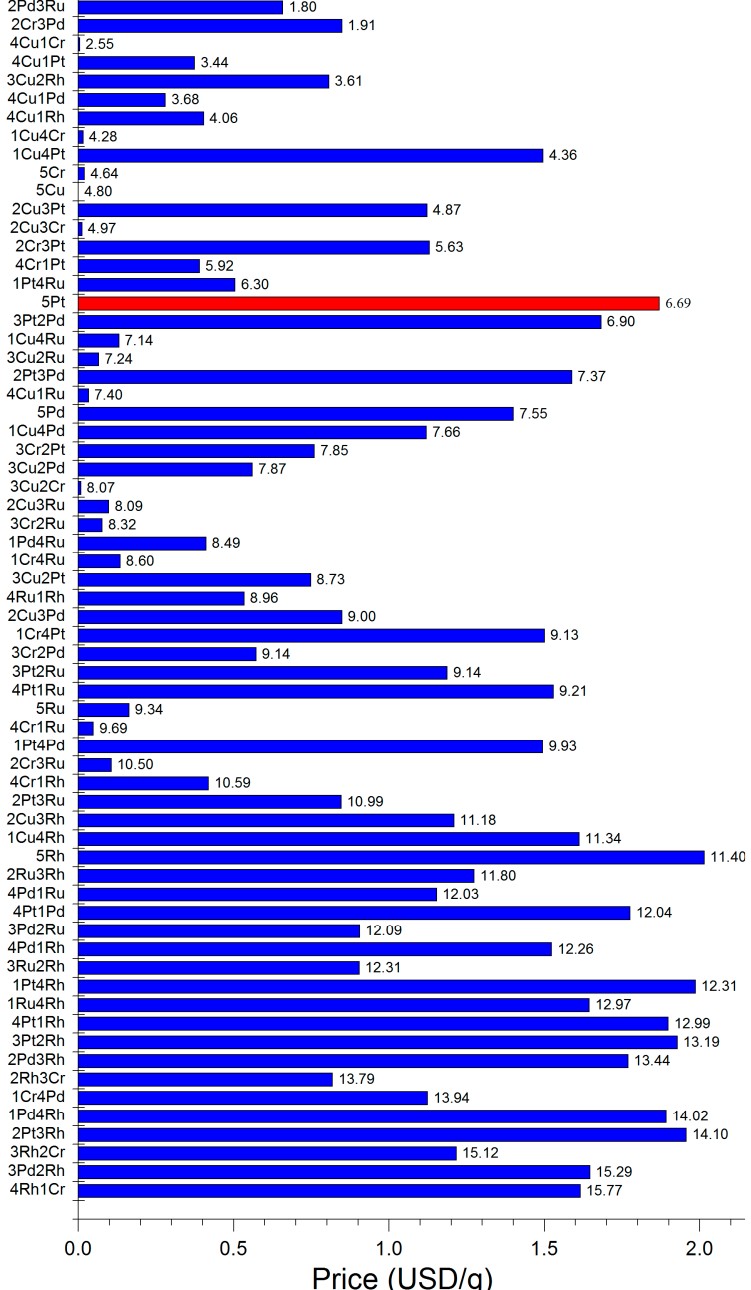

**Figure 1.** Estimated price (USD/g) and % $CH_4$ conversion (value at the end of each bar) of single-metal and bimetallic catalysts. Reaction conditions: feed gas $O_2$/$CH_4$ volume ratio = 3/5, gas hourly space velocity (GHSV) = 15,000 h$^{-1}$, and reactor temperature = 475 °C. Note that the value of $CH_4$ conversion of each catalyst was previously reported by Kumsung et al. [24].

The most interesting catalyst that presented a relatively high $CH_4$ conversion while having a much cheaper cost compared to the benchmark catalyst used combinations of Cr and Ru, especially

2Cr3Ru/Al$_2$O$_3$ and 4Cr1Ru/Al$_2$O$_3$. These two catalysts had CH$_4$ conversion rates of 10.50% and 9.69%, respectively, at an average cost of 0.106 USD/g and 0.048 USD/g, or about 17.6 and 38.9 times cheaper, respectively, than the benchmark catalyst. Therefore, the 2Cr3Ru/Al$_2$O$_3$ catalyst was chosen for further investigations as a representative catalyst of the combinations of Cr and Ru.

The CO and CO$_2$ selectivities, along with the CH$_4$ conversion of 2Cr3Ru/Al$_2$O$_3$ compared with those of 5Cr/Al$_2$O$_3$, 5Ru/Al$_2$O$_3$, and 5Pt/Al$_2$O$_3$, are plotted in Figure 2. Under the same test conditions and the same wt% of active metals loaded on the Al$_2$O$_3$ support for every catalyst, it was observed that 5Cr/Al$_2$O$_3$ had the greatest CO selectivity with the smallest CH$_4$ conversion, while 5Ru/Al$_2$O$_3$ had the greatest CO$_2$ selectivity with the second highest CH$_4$ conversion among these four catalysts. More importantly, 2Cr3Ru/Al$_2$O$_3$ had the greatest CH$_4$ conversion and the greatest CO yield. When the catalysts in Figure 2 were determined in terms of moles of CH$_4$ consumed per total moles of active elements in catalyst per time (TOF), it was found that the TOF values of the 2Cr3Ru/Al$_2$O$_3$, 5Ru/Al$_2$O$_3$, 5Pt/Al$_2$O$_3$, and 5Cr/Al$_2$O$_3$ catalysts were 0.097, 0.153, 0.213, and 0.039 s$^{-1}$, respectively. This revealed that the catalytic activities in terms of TOF were (in order): 5Pt/Al$_2$O$_3$ > 5Ru/Al$_2$O$_3$ > 2Cr3Ru/Al$_2$O$_3$ > 5Cr/Al$_2$O$_3$, which were ranked differently when considering the catalysts in terms of CH$_4$ conversion per total weight of active elements (2Cr3Ru/Al$_2$O$_3$ > 5Ru/Al$_2$O$_3$ > 5Pt/Al$_2$O$_3$ > 5Cr/Al$_2$O$_3$). Consequently, the question arises as to which criteria should be chosen for classifying the catalytic performance. Since the price of catalyst was of greater concern and the price of catalyst per CH$_4$ consumption of the 2Cr3Ru/Al$_2$O$_3$ catalyst was lower than that of the benchmark catalyst under the same testing conditions, the main focus and of particular interest in this work was studying the catalytic activity and the physicochemical properties of the 2Cr3Ru/Al$_2$O$_3$ catalyst, compared to those of each single-metal catalyst component and the benchmark catalyst. Complete characterization, investigation of the operating conditions, and the stability and reusability of the catalyst are presented in Sections 2.2–2.4, respectively.

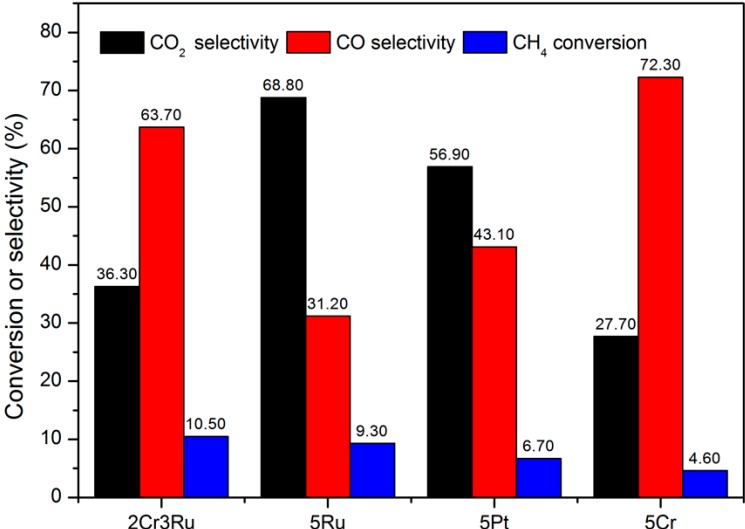

**Figure 2.** Activities of 2Cr3Ru/Al$_2$O$_3$, 5Pt/Al$_2$O$_3$, 5Ru/Al$_2$O$_3$, and 5Cr/Al$_2$O$_3$. Reaction conditions: feed gas O$_2$/CH$_4$ volume ratio = 3/5, GHSV = 15,000 h$^{-1}$, and reactor temperature = 475 °C. The total metal loading of active metals on Al$_2$O$_3$ was fixed at 5 wt%.

## 2.2. Characterization of 2Cr3Ru/Al$_2$O$_3$, 5Cr/Al$_2$O$_3$, 5Ru/Al$_2$O$_3$, and 5Pt/Al$_2$O$_3$

Figure 3 shows the PXRD patterns of the fresh single-metal catalysts 5Pt/Al$_2$O$_3$, 5Ru/Al$_2$O$_3$ and 5Cr/Al$_2$O$_3$, along with a fresh bimetallic 2Cr3Ru/Al$_2$O$_3$ catalyst. The γ-Al$_2$O$_3$ support produced diffraction peaks at 2θ of 32.0, 36.3, 45.8, 60.8, and 67.3 (ICDD 01-080-0956), in all catalysts. It was noticed that the diffraction peaks of γ-Al$_2$O$_3$ were broad or barely visible, indicating that the crystallite size of γ-Al$_2$O$_3$ was small (<7.1 nm, according to Scherrer's equation). The diffraction peaks at 2θ of

39.9, 46.3, and 67.4 (ICDD 00-004-0802) were associated with the metallic Pt. This was of interest as the crystalline phase of the Pt species in the 5Pt/Al$_2$O$_3$ catalyst was not the PtO$_x$ species as expected, because it was calcined in air (750 °C for 6 h). This could be explained by the ability of PtO$_x$ species to decompose at a high calcination temperature, and thus to be present as metallic Pt [31]. The diffraction peaks at 2θ of 28.0, 35.2, 40.2, 54.3, 58.1, 59.5, and 70.0 (ICDD 01-088-0322) were assigned to the RuO$_2$ in the 5Ru/Al$_2$O$_3$ and 2Cr3Ru/Al$_2$O$_3$ catalysts. The diffraction peaks of the Cr component (Cr$_2$O$_3$) in the 5Cr/Al$_2$O$_3$ and 2Cr3Ru/Al$_2$O$_3$ catalysts were undetectable, potentially because it could have been in the amorphous phase or its crystallite size was too small. Comparing the diffraction peaks of RuO$_2$ in the PXRD patterns of the 5Ru/Al$_2$O$_3$ and 2Cr3Ru/Al$_2$O$_3$ catalysts, the intensities of the 5Ru/Al$_2$O$_3$ catalyst were higher than those of the 2Cr3Ru/Al$_2$O$_3$ catalyst. This implied that the amount of crystalline RuO$_2$ in the 5Ru/Al$_2$O$_3$ catalyst was higher than that in the 2Cr3Ru/Al$_2$O$_3$ catalyst, which was in agreement with the weight percentage of Ru in the 2Cr3Ru/Al$_2$O$_3$ catalyst being less than that in the 5Ru/Al$_2$O$_3$ catalyst, as was predetermined during preparation. Furthermore, the crystallite sizes of RuO$_2$ in the 2Cr3Ru/Al$_2$O$_3$ and 5Ru/Al$_2$O$_3$ catalysts (calculated using Scherrer's equation) were 43 nm and 42 nm, respectively; these sizes only changed slightly. This indicated that the presence of Cr in the bimetallic catalyst did not affect the crystallite size of RuO$_2$. More results on the particle sizes of RuO$_2$ and Pt are presented in Figure 5.

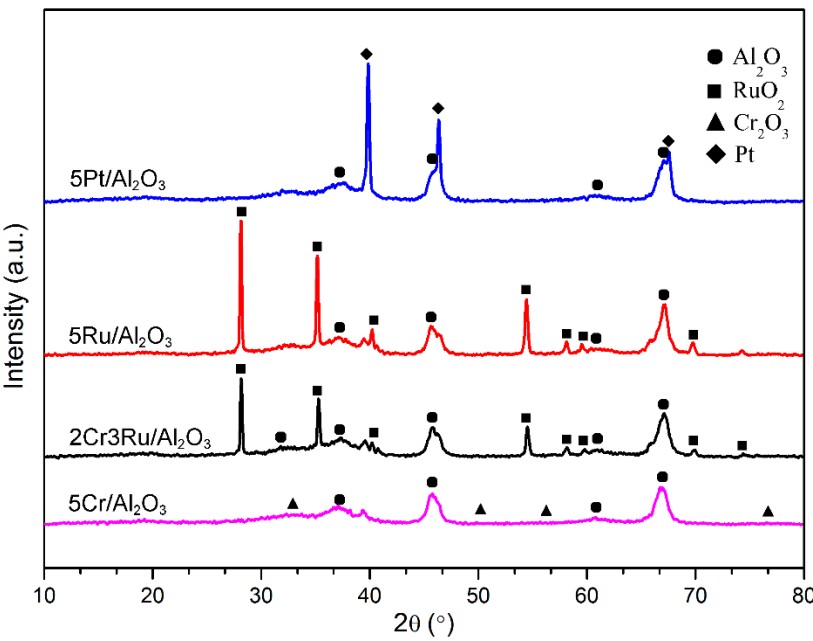

**Figure 3.** PXRD patterns of fresh 5Pt, 5Ru, 5Cr, and 2Cr3Ru supported on Al$_2$O$_3$.

The four selected catalysts were also analyzed using the H$_2$-TPR technique, as shown in Figure 4. It can be seen that the Pt/Al$_2$O$_3$ catalyst did not consume any hydrogen because the Pt was in the metallic Pt(0) phase, as previously found in the PXRD measurement (Figure 3). The 5Cr/Al$_2$O$_3$ catalyst had two broad peaks at around 200 °C and 350 °C, respectively, representing the reductions of Cr$_2$O$_3$ species interacting with the Al$_2$O$_3$ support and the bulk Cr$_2$O$_3$ species [32], respectively. The hydrogen reduction peak of 5Ru/Al$_2$O$_3$ was composed of one large peak with a shoulder peak around 150–250 °C. The lower temperature reduction peak was associated with the RuO$_2$ interacting with the support and the higher temperature reduction peak was assigned to the complete reduction of bulk RuO$_2$ [33,34]. In the presence of RuO$_2$ and Cr$_2$O$_3$ on the Al$_2$O$_3$ support, the hydrogen reduction peak of RuO$_2$ shifted slightly toward a higher temperature (from 180 °C to 200 °C), perhaps because the RuO$_2$ particles had a stronger interaction with the support and/or the Cr$_2$O$_3$ species. It was clear that the reduction peak of Cr$_2$O$_3$ at about 200 °C overlapped with that of the RuO$_2$ species. A small, broad peak also appeared

at around 350 °C, which could be attributed to the bulk $Cr_2O_3$ particles that were influenced by the presence of $RuO_2$.

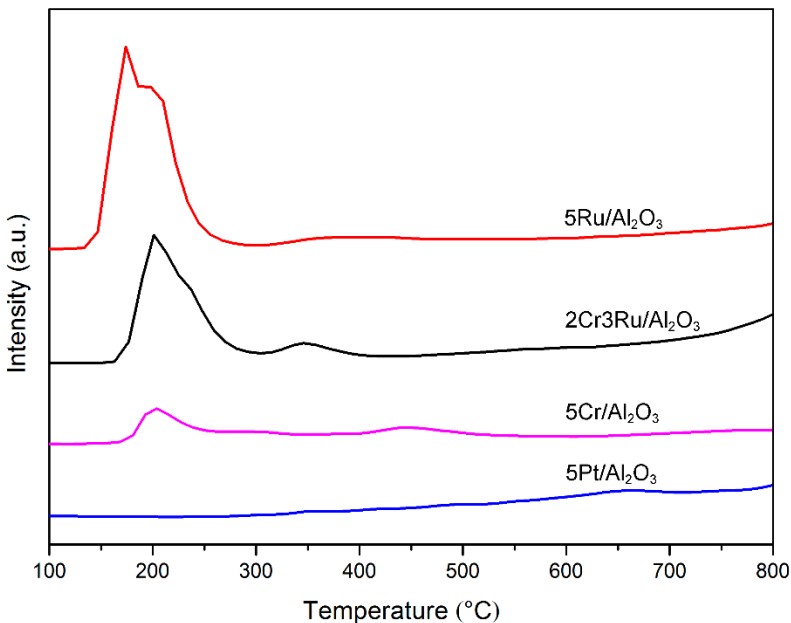

**Figure 4.** $H_2$-TRP profiles of fresh 5Pt, 5Ru, 5Cr, and 2Cr3Ru supported on $Al_2O_3$.

Figure 5 shows the SEM images, SEM/EDS images, and EDS spectra of the fresh $5Pt/Al_2O_3$, $5Ru/Al_2O_3$, $5Cr/Al_2O_3$, and $2Cr3Ru/Al_2O_3$ catalysts. The SEM images for all catalysts were similar; the particles were irregular in shape, and the particle sizes were approximately 50–100 nm, mostly representing the $\gamma$-$Al_2O_3$ support. For all catalysts, the micro-scale distribution of each metal was uniform throughout the catalytic materials (see SEM-EDS image in Figure 5). The EDS spectra of each catalyst confirmed that each element impregnated onto the $\gamma$-$Al_2O_3$ support was present in the materials.

High-resolution TEM images of fresh $5Pt/Al_2O_3$, $5Ru/Al_2O_3$, and $2Cr3Ru/Al_2O_3$ catalysts are shown in Figure 6. Note that the TEM image of the $5Cr/Al_2O_3$ catalyst was also taken, but the $Cr_2O_3$ particles could not be seen (not shown here) because they were in the amorphous phase and could not be distinguished from the alumina particles. The $RuO_2$ particles in the $5Ru/Al_2O_3$ catalyst were mostly in a rutile structure and quite large (approximately 20 nm × 50 nm) with a lattice spacing of 0.25 nm in the $RuO_2$ (1 0 1) plane [35]. Although the $RuO_2$ particles found in the $2Cr3Ru/Al_2O_3$ catalyst were irregular in shape, their fringe distance value of 0.25 nm confirmed that the particles were $RuO_2$ crystals. According to the TEM images, the particles of $RuO_2$ in the $2Cr3Ru/Al_2O_3$ catalyst were approximately 5–10 nm in size, which was smaller than those in the $5Ru/Al_2O_3$ catalyst. This suggested that the addition of $Cr_2O_3$ in the catalyst resulted in a decrease in the average particle size of $RuO_2$, leading to a higher dispersion of active sites. This phenomenon was similar to that reported in a previous study of Ni-Cr bimetallic catalyst for $CO_2$ reforming of $CH_4$, in which Cr was found to inhibit the growth of Ni crystallites, thereby enhancing the dispersion of Ni species in the catalytic material [19]. Hence, we can infer for the Cr-Ru catalytic system that the Cr species inhibited the growth of Ru species, and thus, increased the dispersion of the active Ru species, thereby enhancing the catalyst activity.

With the $5Pt/Al_2O_3$ catalyst, many small dark spots could be seen in the images, corresponding to the Pt particles. The average Pt particle size was approximately 2.5 nm with a lattice spacing of 0.22 nm, which agreed with the (1 1 1) interplanar spacing of Pt [36].

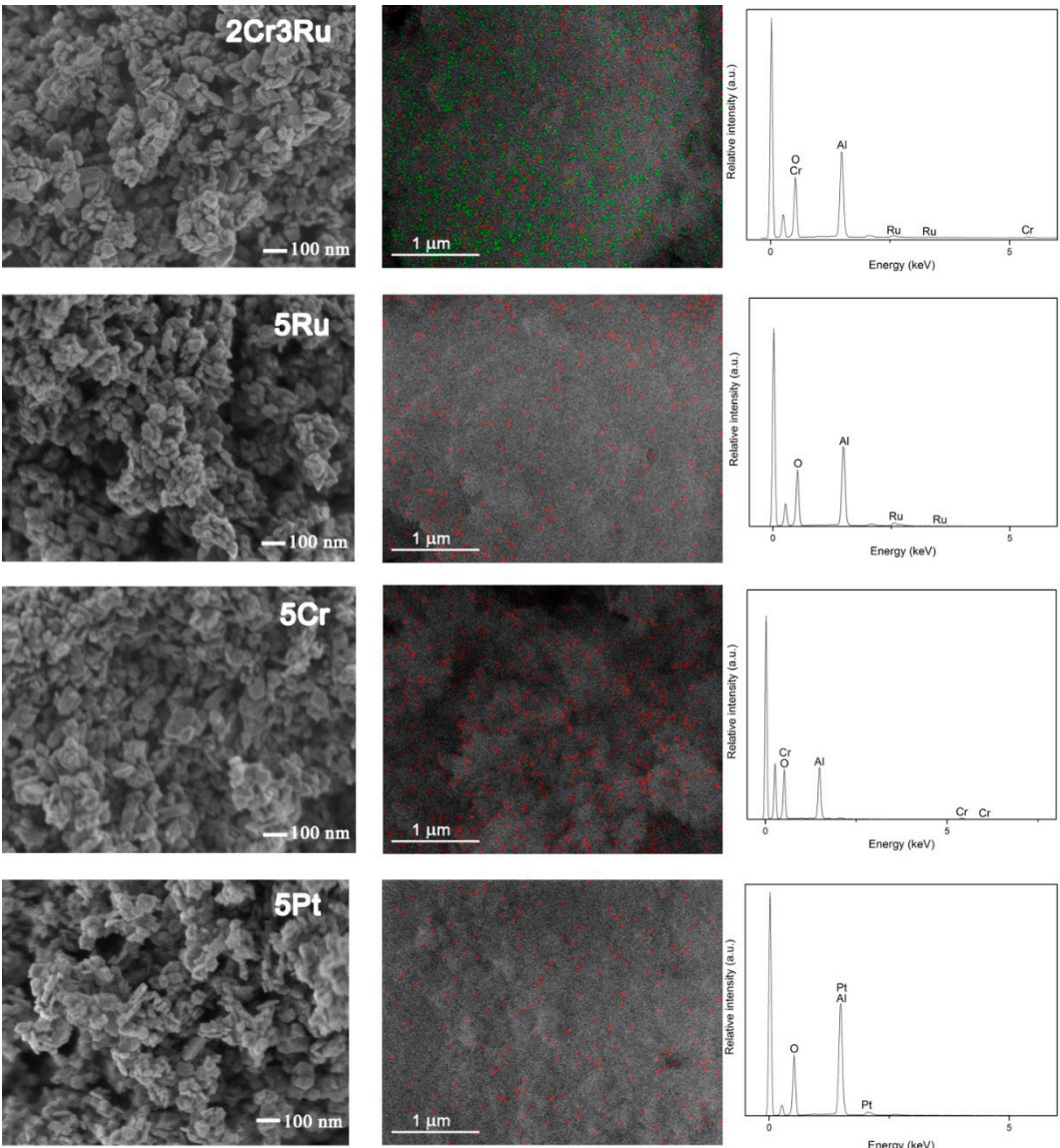

**Figure 5.** SEM images (**left**), SEM/EDS images (**center**), and EDS spectra (**right**) of fresh 5Pt, 5Ru, 5Cr, and 2Cr3Ru supported on $Al_2O_3$.

The binding energies of Cr 2p and Ru 3d for the $5Ru/Al_2O_3$, $5Cr/Al_2O_3$, and $2Cr3Ru/Al_2O_3$ catalysts were examined using XPS, as shown in Figure 7. These spectra confirmed that the catalysts containing Ru consisted of $RuO_2$ (Ru $3d_{5/2}$ = 283.0–283.2 eV, Ru $3d_{3/2}$ = 286.6–287.4 eV) [37], and those containing Cr consisted of $Cr_2O_3$ (Cr $2p_{3/2}$ = 575.6 eV, Cr $2p_{1/2}$ = 585.1 eV) [37]. Of interest was the characteristic XPS peaks of Ru 3d (see Figure 7a) in the $2Cr3Ru/Al_2O_3$ catalyst shifted toward a lower binding energy compared to those of the $5Ru/Al_2O_3$ catalyst. However, the XPS peaks of Cr 2p (see Figure 7b) in the $2Cr3Ru/Al_2O_3$ catalyst remained unchanged relative to those of the $5Cr/Al_2O_3$ catalyst. These results implied that in the presence of $Cr_2O_3$ near the $RuO_2$ particles, not only the $Cr_2O_3$ species inhibited the growth of the $RuO_2$ particles, but the binding energy of Ru 3d also shifted toward a lower energy. This probably resulted in weakening the bond strength of Ru—O. Thus, the mobility of oxygen species involved in the reaction mechanism increased, thereby enhancing the catalytic activity.

Based on the characterization provided using the PXRD, $H_2$-TPR, SEM/EDS, HR-TEM, and XPS techniques combined with the activity results of the four selected catalysts presented in Figure 2, a short summary can be made, as follows. The lowest $CH_4$ conversion was obtained using the $5Cr/Al_2O_3$ catalyst. Evidence showed that the $5Cr/Al_2O_3$ catalyst was in the form of amorphous $Cr_2O_3$.

The active form of the 5Pt/Al$_2$O$_3$ catalyst was crystalline Pt(0), which was in good agreement with other work [38]. The ruthenium species in both the 5Ru/Al$_2$O$_3$ and 2Cr3Ru/Al$_2$O$_3$ catalysts were in the form of crystalline RuO$_2$. Interestingly, the average particle size of RuO$_2$ became smaller in the presence of Cr$_2$O$_3$, and the catalytic activity of the 2Cr3Ru/Al$_2$O$_3$ catalyst was higher than that of the 5Ru/Al$_2$O$_3$ catalyst. This indicated that the addition of Cr$_2$O$_3$ into the RuO$_2$ catalyst could reduce the growth of crystalline RuO$_2$ particles during calcination, resulting in smaller-sized RuO$_2$ crystals, and thus, in an increase in the number of active sites. Moreover, the shift in the Ru 3d binding energy considerably promoted the mobility of oxygen species involved in the reaction, thereby enhancing the reaction.

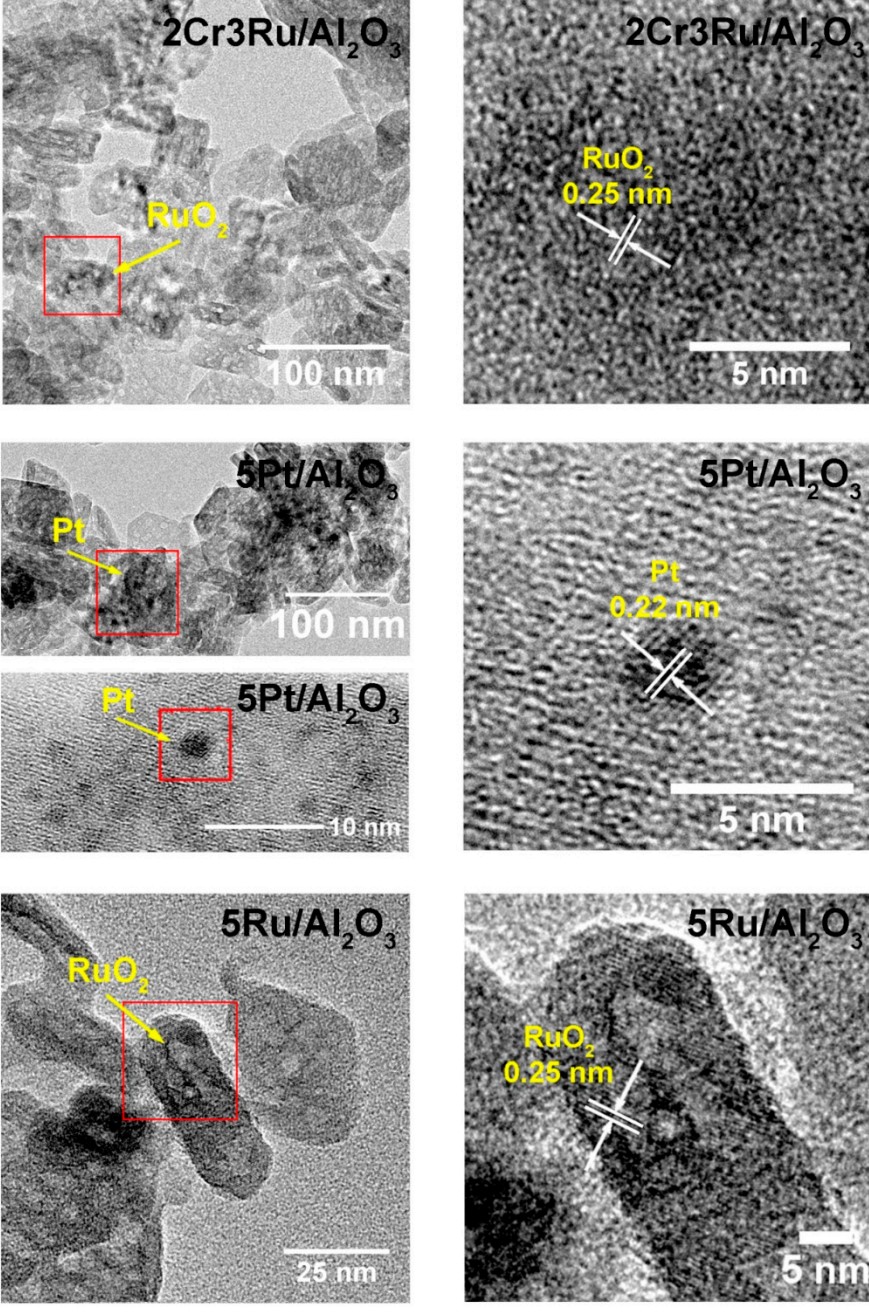

**Figure 6.** HR-TEM images of fresh 5Pt, 5Ru, 5Cr, and 2Cr3Ru supported on Al$_2$O$_3$.

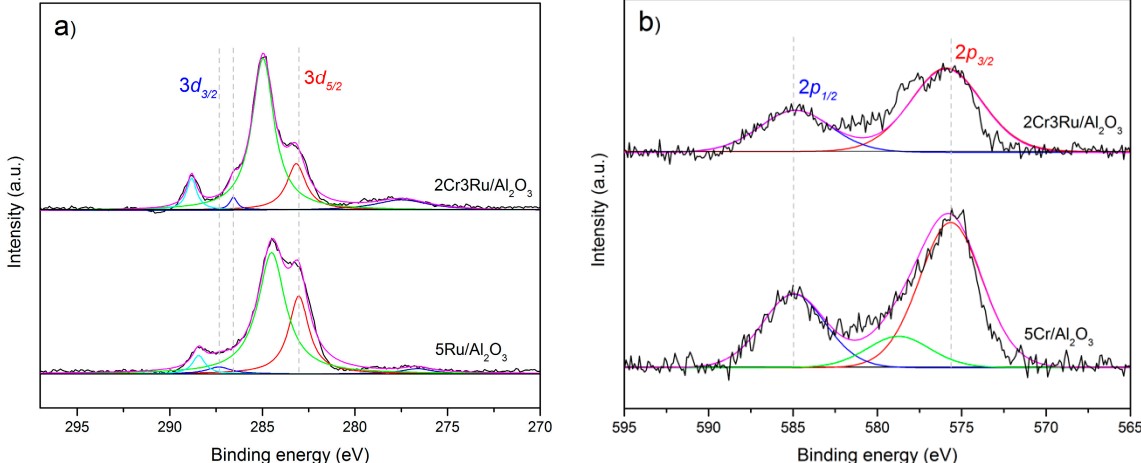

**Figure 7.** XPS spectra of (**a**) Ru 3d and (**b**) Cr 2p regions obtained from 5Ru/Al$_2$O$_3$, 5Cr/Al$_2$O$_3$, and 2Cr3Ru/Al$_2$O$_3$ catalysts.

The NH$_3$-TPD profiles are shown in Figure 8, signifying the acidity of catalyst surfaces of the 5Pt/Al$_2$O$_3$, 5Ru/Al$_2$O$_3$, 5Cr/Al$_2$O$_3$, and 2Cr3Ru/Al$_2$O$_3$ catalysts, along with the bare Al$_2$O$_3$ support. All four catalysts and the Al$_2$O$_3$ support had similar profiles in which a small broad peak around 100 °C was observed, representing NH$_3$ desorption from the weak acidic sites. The Al$_2$O$_3$ support had the lowest number of weak acidic sites. Peaks around 200–350 °C, representing medium acidic sites, were detected for the 5Ru/Al$_2$O$_3$ and 2Cr3Ru/Al$_2$O$_3$ catalysts. The NH$_3$ desorption peak around 450–650 °C, assigned to the strong acidic sites, was seen in the 5Pt/Al$_2$O$_3$, 5Ru/Al$_2$O$_3$, 2Cr3Ru/Al$_2$O$_3$ catalysts. The 2Cr3Ru/Al$_2$O$_3$ catalyst had the maximum intensity at these strong acidic sites, indicating that the additions of active components were able to increase the acidity of the catalysts' surfaces. The results in Figure 1 and the peak areas of acidic sites were compared to relate the methane conversion of each catalyst to the acidity of each catalyst. As can be seen clearly in Figure 1, the CH$_4$ conversions were ranked as: 2Cr3Ru/Al$_2$O$_3$ > 5Ru/Al$_2$O$_3$ > 5Pt/Al$_2$O$_3$ > 5Cr/Al$_2$O$_3$. The analyses of the peak areas of the acidic sites showed that the descending order for the peak areas of the strong acidic sites was: 2Cr3Ru/Al$_2$O$_3$ > 5Ru/Al$_2$O$_3$ ≈ 5Pt/Al$_2$O$_3$ > 5Cr/Al$_2$O$_3$. It was unclear whether only the strong acidic sites or both the strong and medium acidic sites were important for the reaction; because the 5Ru/Al$_2$O$_3$ catalyst was more active than the 5Pt/Al$_2$O$_3$ catalyst, while the peak areas of the strong acidic sites of both catalysts were virtually the same. However, the analysis of the summation of the medium and strong acidic sites revealed that the sum effect of the two sites was (in order) 2Cr3Ru/Al$_2$O$_3$ > 5Ru/Al$_2$O$_3$ > 5Pt/Al$_2$O$_3$ > 5Cr/Al$_2$O$_3$. This suggested that the medium and strong acidic sites, specifically for the strong acidic sites, played important roles in the activation of CH$_4$. It was noted that the weak acidic sites were considered unimportant active sites since the 5Cr/Al$_2$O$_3$ and the Al$_2$O$_3$ supports were not highly active in the reaction.

In general, the activation of C–H bonds in CH$_4$ does not take place easily without sufficient energy being available. In the absence of a catalyst, CH$_4$ and O$_2$ can react to produce various products (such as CO$_x$, ethane, ethylene, and propane) at high temperatures (>700 °C) and atmospheric pressure. However, in the presence of a solid catalyst, C–H bond activation can take place under mild conditions. In some previous studies, catalysts that exhibited a large number of Brønsted acid sites (strong acidic sites) had a strong effect on the activation of C–H bond cleavage of CH$_4$. Examples of such catalysts with strong acidic surfaces are Pd/ZSM-5 [39] and Au-Pd/nanohybrid 3D macroporous La$_{0.6}$Sr$_{0.4}$MnO$_3$ [22]. Catalysts like these may have analogous behavior to the 2Cr3Ru/Al$_2$O$_3$ catalyst, where the strong acidic sites are the active sites and there is initially cleavage of the C–H bonds after the CH$_4$ molecule have adsorbed on the catalyst surface. Then, the intermediate species can further produce CO$_x$ and H$_2$O. It should be noted that this mechanism can create oxygen vacancies due to the loss of adsorbed

oxygen from the metal oxides or the support. The oxygen from the gas phase subsequently refills the empty sites before starting a new cycle [40].

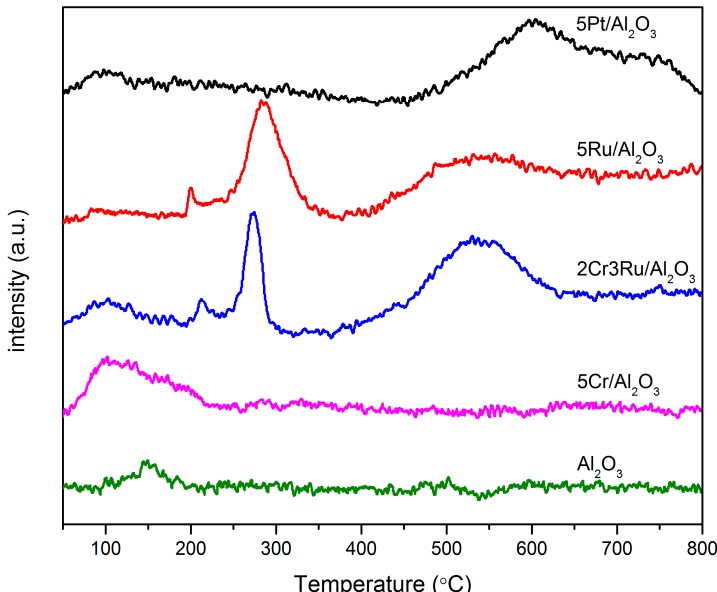

**Figure 8.** $NH_3$-TPD profiles of fresh 5Pt, 5Ru, 5Cr, and 2Cr3Ru supported on $Al_2O_3$, along with bare $\gamma$-$Al_2O_3$.

### 2.3. Investigation of Operating Conditions (Reactor Temperature and $O_2$/$CH_4$ Ratio)

The effect of reactor temperature on the activity of the 5Pt/$Al_2O_3$, 5Ru/$Al_2O_3$, 5Cr/$Al_2O_3$, and 2Cr3Ru/$Al_2O_3$ catalysts is shown in Figure 9, where the catalytic performance of each catalyst is presented in terms of $CH_4$ conversion versus reactor temperature and TOF versus reactor temperature. In Figure 9a), the 2Cr3Ru/$Al_2O_3$ catalyst clearly achieved the highest $CH_4$ conversion at every temperature above 300 °C. In particular, the light off-temperature of the 2Cr3Ru/$Al_2O_3$ catalyst was established at the lowest temperature, approximately 50 °C and 100 °C lower than the light off-temperatures of the 5Pt/$Al_2O_3$ and 5Ru/$Al_2O_3$ catalysts, respectively. For the 5Cr/$Al_2O_3$ catalyst, the $CH_4$ conversion slowly increased, and there was no clear light-off temperature. However, the catalytic performance of the 5Cr/$Al_2O_3$ catalyst was much lower than that of the 5Pt/$Al_2O_3$, 5Ru/$Al_2O_3$, and 2Cr3Ru/$Al_2O_3$ catalysts for reactor temperatures above 400 °C. In Figure 9b), it can be clearly seen that the TOF values of the 5Pt/$Al_2O_3$ catalyst were higher than those of the other three catalysts. Above 400 °C, the order of the TOF values for every temperature was: 5Pt/$Al_2O_3$ > 5Ru/$Al_2O_3$ > 2Cr3Ru/$Al_2O_3$ > 5Cr/$Al_2O_3$. In fact, this indicated that one active site of the 5Pt/$Al_2O_3$ catalyst was the most active among the others. However, from the catalyst price viewpoint and $CH_4$ consumption, the 2Cr3Ru/$Al_2O_3$ catalyst was more economical than the 5Pt/$Al_2O_3$ and 5Ru/$Al_2O_3$ catalysts.

The effect of varying the $O_2$/$CH_4$ ratio is presented in Figure 10. The experiment was performed to investigate the performance of the 2Cr3Ru/$Al_2O_3$ catalyst under fuel-rich and stoichiometric conditions. The reactor temperature was set at 475 °C, and atmospheric pressure was used. When the $O_2$/$CH_4$ ratio increased from 3/5 to 2/1 (0.6 to 2.0), the $CH_4$ conversion steadily increased from about 10% to 75%. Similarly, the $CO_2$ selectivity progressively increased from about 27% to 100%, which was the opposite to the trend with the CO selectivity, which decreased from 73% to 0%. These results indicated that the amount of $O_2$ an elevate the limit of $CH_4$ conversion. Basically, complete combustion (when the burning process produces only $CO_2$ and $H_2O$) is highly favorable when there is a plentiful supply of oxygen. This showed that no CO is produced when using this catalyst in the stoichiometric or fuel-lean conditions. In contrast, incomplete combustion (involving the partial combustion of $CH_4$)

occurs when there is insufficient oxygen supply to use all the fuel (fuel-rich condition), and thus CO is preferentially produced.

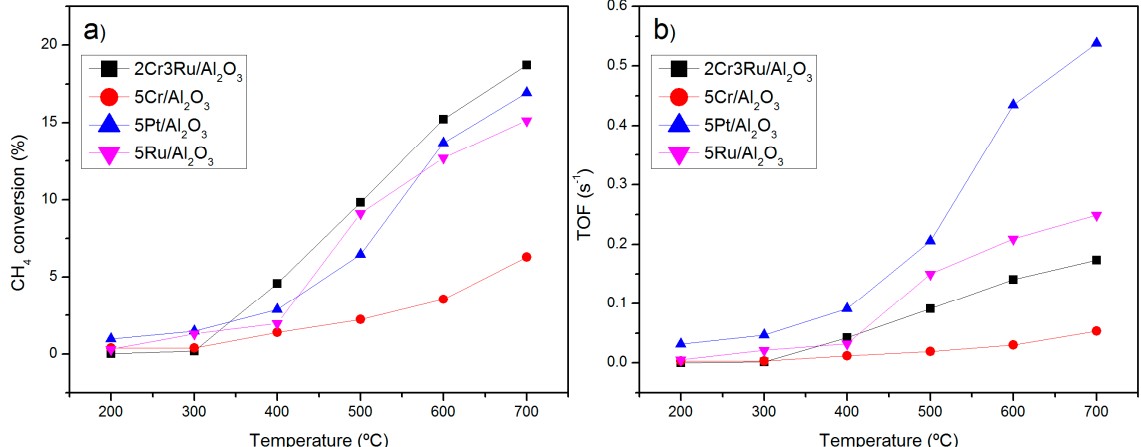

**Figure 9.** Effect of varying reactor temperature on activity of 5Pt, 5Ru, 5Cr, and 2Cr3Ru supported on $Al_2O_3$. (**a**) Plots of $CH_4$ conversion versus reactor temperature and (**b**) plots of TOF versus reactor temperature. Reaction conditions: feed gas $O_2/CH_4$ volume ratio = 3/5, GHSV = 15,000 $h^{-1}$, and reactor temperature = 200–700 °C. Note that an experimental error was ±1.5%, calculated from nine tests of 2Cr3Ru/$Al_2O_3$ under the same conditions at 475 °C.

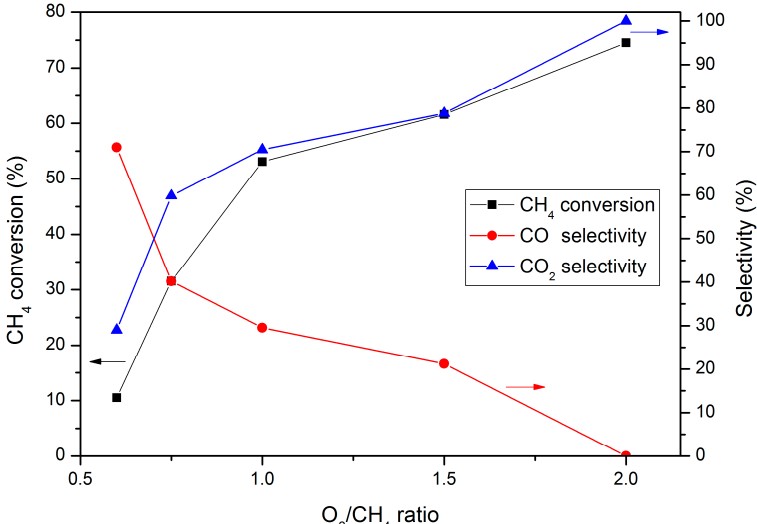

**Figure 10.** Effect of varying $O_2/CH_4$ ratio on activity of 2Cr3Ru/$Al_2O_3$ catalyst. Reaction conditions: feed gas $O_2/CH_4$ volume ratio = 3/5–2/1 or 0.6–2.0, GHSV = 15,000 $h^{-1}$, and reactor temperature = 475 °C.

## 2.4. Investigation of Stability and Reusability of 2Cr3Ru/$Al_2O_3$ Catalyst

Catalyst stability is essential when using the catalyst for many hours. To monitor whether the catalyst could maintain its activity for several hours, a time-on-stream experiment of the 2Cr3Ru/$Al_2O_3$ catalyst was performed, and Figure 11 shows the $CH_4$ conversion, CO selectivity, and $CO_2$ selectivity attained during 24 h of continuous operation at 475 °C and a volumetric $O_2/CH_4$ ratio of 3/5 using a GHSV of 15,000 $h^{-1}$. It can be seen that the $CH_4$ conversion started at 5.6% in the first hour and then gradually increased to 11% by the fourth hour. After that, it minimally decreased to approximately 10% and remained steady at about 10% until the last hour of operation. The CO selectivity gradually decreased during the first 8 h, from 75% to 60% and then remained virtually unchanged during the remainder of the test period. In contrast, the $CO_2$ selectivity slowly increased from 25% to 40% in 8 h, and then remained constant at about 40%. This suggested that the 2Cr3Ru/$Al_2O_3$ catalyst was

highly robust for the reaction. As previously mentioned, the CO selectivity was greater than the $CO_2$ selectivity, since the generation of CO is undoubtedly favored under the fuel-rich condition, due to insufficient $O_2$ for complete combustion.

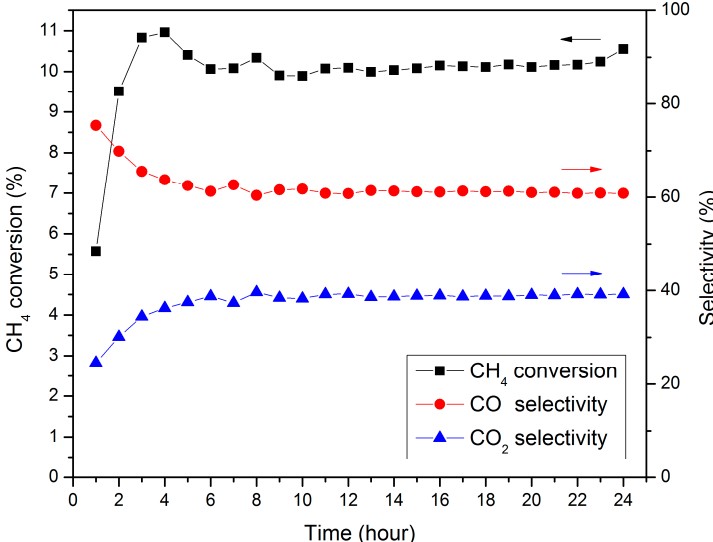

**Figure 11.** Time-on-stream testing results of 2Cr3Ru/Al$_2$O$_3$ catalyst during 24 h. Reaction conditions: feed gas $O_2$/$CH_4$ volume ratio = 3/5, GHSV = 15,000 h$^{-1}$, and reactor temperature = 475 °C.

To examine whether coke deposition occurred when the catalyst was used for 24 h, fresh and used 2Cr3Ru/Al$_2$O$_3$ catalysts were analyzed using TG-DTA, as shown in Figure 12. Only one clear DTA peak around 97 °C was observed for both catalysts, ascribed to the loss of water [41]. No clear TG-DTA peak of coke decomposition was seen—TG-DTA peaks of the coke decomposition are normally seen around 500–550 °C [42]—indicating that coke deposition did not occur during the course of testing. Additionally, no black film or black fine powder was observed on the reactor tube's surface or the catalyst (data not shown here), confirming that the catalyst did not have any problems with coke deposition.

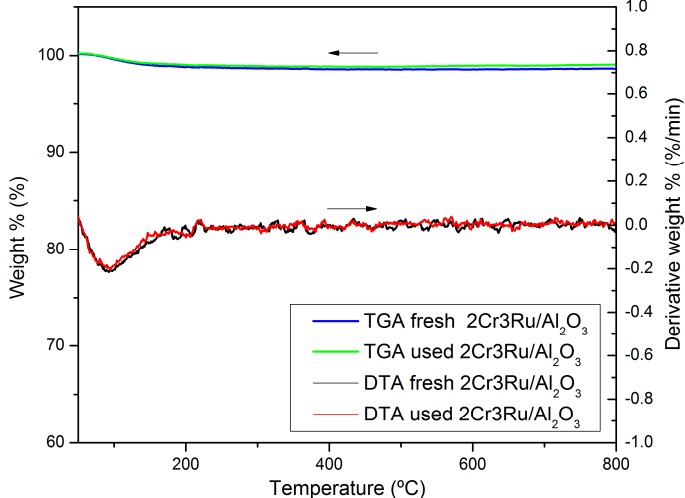

**Figure 12.** TG-DTA curves of fresh and used 2Cr3Ru/Al$_2$O$_3$ catalysts.

One of the main factors for consideration in using an industrial catalyst is its reusability. Figure 13 presents the results of reusability testing with 2Cr3Ru/Al$_2$O$_3$ for five cycles. Each cycle was carried out by running the normal testing conditions for 6 h. Then, the reactor tube containing the catalyst was

removed from the testing system and left to stand in air at room temperature for about 18 h. After that, a new testing cycle using that same catalyst was begun. The purpose of this method was to simulate the conditions when the catalyst was used multiple times without regeneration. The cycle-averaged values of $CH_4$ conversion from cycle #1 to cycle #5 were 9.5%, 10.5%, 10.3%, 10.5%, and 10.4%, respectively, but the last 3 h of each cycle had about the same average $CH_4$ conversion (10.4%). Promisingly, the averages of $CH_4$ conversion after the first cycle (for cycles #2–5) were close to 10.4%, suggesting that the 2Cr3Ru/Al$_2$O$_3$ catalyst was quite durable and could be used multiple times. It should be noted that the activity of the catalyst during the first 2 h was substantially lower than the average methane conversion, perhaps because the catalyst needed time to reach equilibrium and/or the catalyst's surface was not fully activated due to surface impurities (moisture, hydroxyl forms). Moreover, the methane conversion for the first 2 h in cycle #1 was rather lower than for cycles #2–5. This implied that the surface of the fresh catalyst was not as active as the used catalyst, possibly because the time that the fresh catalyst was left standing in air before testing was much longer than the per-cycle time left standing in air for the used catalyst (the fresh catalyst after calcination was left standing in air for 2 days before testing and the used catalyst was left standing in air for about 18 h before testing in each cycle), so that moisture or the hydroxyl form was able to cover more of the surface of the fresh catalyst than that of the used catalyst.

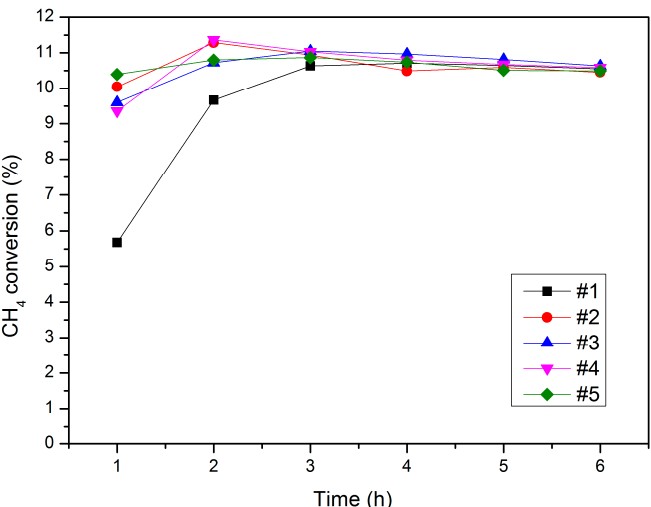

**Figure 13.** Reusability test of 2Cr3Ru/Al$_2$O$_3$ catalyst during five cycles (#1–#5). Reaction conditions: feed gas O$_2$/CH$_4$ volume ratio = 3/5, GHSV = 15,000 h$^{-1}$, and reactor temperature = 475 °C.

## 3. Materials and Methods

### 3.1. Catalyst Preparation

All the single and bimetallic catalysts were prepared using wet impregnation as previously reported [24]. For example, 2 wt% Cr + 3 wt% Ru loaded onto the $\gamma$-Al$_2$O$_3$ support (denoted as the 2Cr3Ru/Al$_2$O$_3$ catalyst) was prepared as follows. Cr(NO$_3$)·9H$_2$O (98.5% purity, Alfa Aesar, Massachusetts, MA, USA) and RuCl$_3$ (Ru 47.7 min, Alfa Aesar) were dissolved in deionized water to have a molarity of Cr ion solution = 0.485 M and Ru ion solution = 0.671 M, respectively. Amounts of Cr 2 mg and Ru 3 mg were determined from the stock solutions and pipetted into 95 mg of the $\gamma$-Al$_2$O$_3$ support (Alfa Aesar, 99.97% purity) that was contained in a ceramic crucible. A magnetic bar was added into the mixture which was stirred using a heater-stirrer (Biobase, MS7-H550-Pro, Shandong, China) at 500 rpm and room temperature for 1 h and then heated to 165 °C and stirring continued until the mixture had dried. A fine powder was obtained and then taken to calcine at 750 °C for 6 h in an air furnace (Kejia furnace, KJ-1600C, Zhengzhou, China) with a heating rate of 10 °C/min. After naturally cooling down to room temperature, the 2Cr3Ru catalyst was obtained. Similarly, 5 wt% Cr, 5 wt%

Ru, and 5 wt% Pt ($PtCl_2$, Alfa Aesar, 98% purity, Pt ion solution = 0.039 M) loaded onto the $\gamma$-$Al_2O_3$ support (denoted as 5Cr/$Al_2O_3$, 5Ru/$Al_2O_3$, and 5Pt/$Al_2O_3$, respectively) were prepared using the same procedure.

### 3.2. Catalyst Activity Test

A sample of 10 mg of the catalyst was tested for the partial combustion of $CH_4$ over a temperature range of 200–700 °C and at atmospheric pressure in a continuous tubular plug flow reactor (Kejia furnace, KJ-O1200-60IC, Zhengzhou, China). The catalyst was packed in a quartz tube (0.5 cm inside diameter) and sandwiched between quartz wool. Then, a feed gas consisting of $O_2$ (Linde, 99.97% purity, Samutprakarn, Thailand) and $CH_4$ (Praxair, 99.999% purity, Bangkok, Thailand) at a volume ratio of $O_2$/$CH_4$ of 3:5, without any inert gas, was fed into the tubular reactor at a total feed flow rate of 50 mL/min, corresponding to a gas hourly space velocity (GHSV) of 15,000 $h^{-1}$. The $O_2$/$CH_4$ ratio of the feed gas was also varied from 0.6–2.0. All the feed gas flow rates were controlled using mass flow controllers (KOFLOC 3810 DSII, Hiatt Tamarind Company Ltd., Bangkok, Thailand). The effluent gas was analyzed using a gas chromatograph (GC, SHIMADZU, GC-14A, Kyoto, Japan) equipped with a thermal conductivity detector (TCD). Typically, the activity was evaluated after the system reached steady state at 3 h after the reactor temperature had reached the set point. Only $CO_2$, CO, $H_2$, and $H_2O$ were detected as products. The activity of the catalyst was calculated based on the carbon balance of the gaseous species and expressed in terms of the amount of the $CH_4$ conversion, CO selectivity, and $CO_2$ selectivity, as shown in Equations (1)–(3), respectively:

$$\%CH_4 \text{ conversion} = 100 \times \frac{CH_{4in} - CH_{4out}}{CH_{4in}} \tag{1}$$

$$\%CO_2 \text{ selectivity} = 100 \times \frac{\text{moles of } CO_2}{CH_4 \text{ consumed}} \tag{2}$$

$$\%CO \text{ selectivity} = 100 \times \frac{\text{moles of } CO}{CH_4 \text{ consumed}} \tag{3}$$

The TOF values of the catalysts were defined as moles of $CH_4$ consumed per total moles of active elements in catalyst per time. In addition, a time-on-stream experiment was carried out for 24 h to monitor the reaction of the products. A reusability test of the catalyst was performed by repeating these steps for five cycles: running the catalyst testing system for 6 h, removing the reactor tube that contained the catalyst from the testing system, letting the catalyst sit in air at room temperature for 18 h, and then re-testing the catalyst.

### 3.3. Catalyst Characterization

A powder X-ray diffractometer (PXRD, JEOL JDX-3530 and Philips X-Pert, Tokyo, Japan, using Cu-$K_\alpha$ radiation, 45 kV, and 40 mA, 0.02° step size, 0.5 s time per step, 2θ range from 10° to 80°) was used to obtain PXRD patterns of the samples.

The $H_2$-temperature programmed reduction ($H_2$-TPR) technique was used to determine metal-support and metal-metal interactions. A sample of 50 mg was used. The measurements were carried out to achieve $H_2$-TPR profiles by operating in a continuous-flow Inconel tube reactor held at 25–800 °C with a heating rate of 5 °C/min. An $H_2$/Ar mixture gas (9.6% $H_2$) was introduced into the catalyst bed at a total flow rate of 30 mL/min. The $H_2$ consumption was continuously monitored using a TCD-equipped GC (Shimadzu GC-2014, Shimadzu Corporation, Kyoto, Japan).

A scanning electron microscope with an energy dispersive X-ray spectrometer (SEM/EDS; FE-SEM: JEOL JSM7600F, JEOL Ltd., Tokyo, Japan, 3.9 mm working distance) was used to obtain images of the surface morphology and element distribution of the samples. Every sample was coated with gold using a gold sputtering technique before the characterization.

A high-resolution transmission electron microscope (HR-TEM; JEOL JEM-3100F, JEOL Ltd., Tokyo, Japan, operated at 300 kV) was used to image particles at the nano-scale and to analyze particle sizes of the samples. A small amount of each sample was mixed with ethanol and dispersed using sonication. The solution was dropped on a carbon-coated Cu grid and dried at room temperature.

The binding energies of Ru 3d and Cr 2p for the catalysts were characterized using X-ray photoelectron spectrometry (XPS, Kratos Axis Ultra DLD, Kratos Analytical Ltd., Manchester, UK, using Al Kα for the X-ray source).

An ammonia-temperature programmed desorption ($NH_3$-TPD; TPD/R/O Thermo Finnigan 1100, Thermo Finnigan LLC, San Jose, CA, USA) technique was used to measure the surface acidity of the catalytic materials. In brief, the catalysts were pretreated under He flow at 400 °C for 1 h and cooled to 40 °C before 10% $NH_3$/He mixed gas was allowed to flow over the catalysts for 30 min to adsorb on the acid sites. Any excess ammonia was eradicated by flowing $N_2$ at 40 °C for 20 min. The catalysts were then heated to 800 °C at a heating rate of 10 °C/min while He was passed over the catalysts at 30 mL/min. A TCD detector was used to obtain the $NH_3$-TPD profiles.

A thermal gravimetrical and differential temperature analyzer (TG-DTA, PerkinElmer Pyris 1 TGA, PerkinElmer Limited, Bangkok, Thailand) measured the thermal decomposition of the catalyst and the difference in temperature between the sample and the reference. A sample of 15 mg of the catalyst was loaded into an alumina crucible and heated under a flow of air (99.99% purity, Thai Standard Gas (TSG) CO., Ltd., Bangkok, Thailand) at a flow rate of 100 mL/min from room temperature to 800 °C with a heating rate of 10 °C/min.

## 4. Conclusions

The cost analysis of the six most active single-metal catalysts and 60 bimetallic catalysts, prepared from combinations of the six metals using different metal ratios, revealed that the $2Cr3Ru/Al_2O_3$ catalyst had the highest $CH_4$ conversion at a cost substantially lower than that of the benchmark catalyst (5 wt% of Pt on $\gamma$-$Al_2O_3$). The estimated cost of the $2Cr3Ru/Al_2O_3$ catalyst was 0.106 USD/g, which was about 17.6 times cheaper than the benchmark catalyst. PXRD and $H_2$-TPR measurements revealed that the Cr and Ru components were in the forms of amorphous $Cr_2O_3$ and crystalline $RuO_2$. Based on the HR-TEM images, the presence of $Cr_2O_3$ reduced the particle size of the crystalline $RuO_2$, thereby increasing the dispersion of the $RuO_2$ particles, and thus resulting in enhanced $CH_4$ conversion. The XPS measurements showed that the addition of $Cr_2O_3$ into $RuO_2$ resulted in a shift of the characteristic peak of Ru 3d toward a lower binding energy, considerably enhancing the reaction by weakening the bond strength of Ru—O and thus improving the oxygen mobility during the reaction mechanism. Moreover, the characterization of the $2Cr3Ru/Al_2O_3$ catalyst using $NH_3$-TPD showed that the medium and strong acid sites played important roles in the activation of $CH_4$. The investigation of the reactor temperature and the $O_2$/$CH_4$ ratio suggested that $CH_4$ conversion and the CO and $CO_2$ selectivities were strongly influenced by the operating conditions. A time-on-stream test carried out over 24 h and a reusability test of five cycles yielded promising results. The difference in the costs of the $2Cr3Ru/Al_2O_3$ and $Pt/Al_2O_3$ catalysts was estimated, and activity tests at different reactor temperatures using the $2Cr3Ru/Al_2O_3$ catalyst were compared to those of the $Pt/Al_2O_3$ catalyst. The results indicated that the $2Cr3Ru/Al_2O_3$ catalyst was more economical than the $Pt/Al_2O_3$ catalyst, although the TOF values of the latter were actually more reactive than those of the former.

**Author Contributions:** Conceptualization, A.S. and S.S.; Methodology, T.C., W.K. and A.S.; Data Curation, T.C.; Writing—original draft preparation, T.C. and A.S. Writing—review and editing, A.S. and M.C.

**Funding:** This research was funded by the Kasetsart University Research and Development Institute (KURDI), Bangkok, Thailand, the Center of Excellence on Petrochemical and Materials Technology, the National Nanotechnology Center (NANOTEC), NSTDA, Ministry of Science and Technology, Thailand, through its program of Research Network NANOTEC (RNN), the Thailand Research Fund (TRF) grant number IRG5980004), and the Commission on Higher Education. T.C. received funding through a scholarship from the Department of Chemical Engineering at Kasetsart University.

**Conflicts of Interest:** The authors declare no conflict of interest.

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
