# Peer review of "Chromium-Ruthenium Oxides Supported on Gamma-Alumina as an Alternative Catalyst for Partial Combustion of Methane"

_catalysts, doi:10.3390/catal9040335_

Round 1

Reviewer 1 Report

In this work, partial combustion of methane has been investigated over Cr-Ru catalysts supported on γ-Al2O3, which was proposed as a valid alternative to the Pt/ γ-Al2O3 benchmark catalyst. The paper provide interesting results; however, some improvements are required before recommending it to be accepted for a publication on Catalysts.

1.      Figure 1 compare the price and the mean CH4 conversion of several catalysts employed for methane partial oxidation. However, it is mandatory to specify the operative conditions (temperature, space velocity, feeding ratios and so on) for every test; in fact, the methane conversion cannot be an absolute value but it strongly depends on the selected operative conditions. After this evaluation, very different conclusions could be drawn.

2.      Please provide the operative conditions in the caption of Figure 2.

3.      Different authors have reported that the addition of a second metal in the bimetallic formulations increases active species dispersion. Please, provide a comment on this (See for example Palma et al., Catalysts 2017, 7(8), 226; https://doi.org/10.3390/catal7080226).

4.      The results of XRD and TPR analysis revealed that Pt was presented in a metallic state on the catalyst. However, the sample was prepared by calcination in air. Please provide a detailed description of this phenomenon.

5.      The authors stated that the catalyst was left standing in air before stability test and after each cycle. Please, provide an explanation for this choice.

Author Response

Dear Reviewer,

Please see the file attached for the responses to the comments.

Best Regards,

Reviewer 2 Report

Reviewing of the manuscript: Chromium-ruthenium oxides supported on gamma-alumina as an alternative catalyst for partial combustion of methane

A Series of single-metal and bimetallic catalysts were evaluated in the partial combustion of methane and compared with the benchmark Pt/gamma-Al2O3 system, and some of them presented a superior performance than that of the commercial one. In addition to the catalytic activity evaluation, a cost analysis revealed that the (2wt%Cr + 3wt%Ru)/gamma-Al2O3 system presents not only a superior catalytic activity than the commercial catalyst but also it is 17.6 times cheaper. Besides this, based on further experiments the authors suggested that the 2%Cr+3%Ru/gamma-Al2O3 system presents a certain catalytic stability that makes it a suitable catalyst for commercial uses.

In general, the manuscript presents a surface description of a series of results (characterization and catalytic activity measurements) and provides an attempt to compare the prepared systems with a commercial catalyst. However, the work is weak because the main purpose, which is the comparison of the catalysts is based on inappropriate criteria. Just the conversion or the selectivities are not criteria for selecting the best catalyst from a series of catalysts. Additionally, the characterization carried out over the prepared materials is poorly discussed. It is just descriptive. Moreover, there was not a correlation of the information obtained from the different techniques to better profile the selection of the catalyst.

Therefore, I do not recommend the publication of this manuscript in the journal Catalysts.

More specific comments are presented below.

Although the same protocol was followed for preparing all the home-made materials, the homogeneity of the obtained materials could be questionable. During the drying of mixture (support + metal precursors solution), the preferential deposition over some portions of the materials is hardly avoidable. Especially if the conditions of heat of stirring (method of stirring) were not strictly controlled. The authors should provide a more detailed description of the control of all the parameters involved during the synthesis.

As for the catalytic activity measurement, the amount of sample evaluated per test was so few (10 mg), and considerable errors on determining the amount of loaded catalyst is expectable. Therefore, small differences in the amount of loaded catalyst may result in important alterations of the catalytic performance. Furthermore, how the authors ensure that the evidence of heat and mass transport phenomena has been minimized? This is important, especially if an accurate comparison between the catalytic performance is being proposed.

How was the temperature controlled during the catalytic activity tests?

Regarding the time on stream experiment, 14 h is a short period to suggest that the catalyst is stable under reaction conditions.

Within the description of the characterization techniques, further details have to be provided. Such in the case of the XRD experiments: step size? Time per step?

In the TPR, how much sample was loaded?

Regarding the results and discussion, the Figure 1 should be explained. Have the systems included in that figure been analyzed in this work or was this carried out in other paper? This should be clarified at least in the Figure caption.

Which is the criteria for obtaining the conversion and selectivity presented in Figure 2? Are these values obtained at the same temperature in all cases? A more normalized criteria such as the TOF should be applied in order to compare different catalysts. Especially if different metal dispersion can be expected due to the different formulations of the catalysts. The basis for the comparison of catalysts performance seems to be unsuitable.

Within the XRD analysis, the authors suggested that the presence of Cr in bimetallic catalysts does not affect the crystallite size of RuO2. However, what about the availability of Ru species? The alterations of the reflection lines depending on the catalyst formulation is clearly demonstrating modifications in the size of the metal clusters and consequently the metal dispersion should be different. Therefore, the catalytic activity may be also different and this criterion was not considered within the comparison of the catalytic performance.

Moreover, if the acidic properties of the catalysts are also determinant in their performance, how was this considered during the comparison?

In Figure 8, where the catalytic activity of the different catalysts is depicted against the temperature, the conversion values are low. Therefore, what about the error of these experiments? The differences between the commercial catalysts and the system highlighted by the authors how the most active, are below 2% almost 3%, thus are this enough to establish a considerable difference between the catalysts?

Author Response

(The authors gave the same response as above.)

Reviewer 3 Report

The paper by Chomboom et al. reports the catalytic activity of series of CrOx, RuOx and RuOx-CrOx catalysts supported on alumina in the partial oxidation of methane. The activity of these materials is compared to that of a benchmark Pt/Al2O3 system. The paper introduces some catalysts costing analysis which are interesting. The overall idea of the work could be appealing for the catalysis community however there are some serious drawbacks that the need to be addressed before publication.

1)      The authors tested the catalysts’ stability using semi-long term runs (indeed 6 hours is a very short run) and recyclability tests. In order for this data to be meaningful the equilibrium conversion should be calculated and included in the graph. The possibility of running these tests under close to equilibrium conversion could be masking the catalysts deactivation. Basically we need to be sure that we are running in the kinetic regime and we are not under thermodynamic control.

2)      Post reaction characterisation is a big missing of this paper. Is there any means of carbon deposition or metal oxides sintering? Methane decomposition typically leads to carbon deposits that deactivates the catalysts. This referee would like to see a post reaction Raman, TPO and or/TGA and compared the carbon deposition of the studied catalysts (which could be also correlated to the acidity study discussed in this work)

3)      The synergistic effect RuOx-CrOx is very interesting indeed. Considering that catalysis is a surface phenomenon and that the electronic interaction Ru-Cr is very relevant in this multicomponent system, it is strongly recommended to conduct a XPS study on the prepared materials. XPS data will clarify the electronic interactions and will shed some lights on the synergistic effect beyond acidity and particle size distribution. In fact, electronic effects can also explain the capacity of a catalytic surface to interact and activate the reactive molecules. Such a study is key to explain the selectivity trends.

4)      As for the selectivity – have the authors calculate the Carbon balance for this processes? It seems that they are closing the balance very close to 100% (but never at 100%) in view of the selectivity trends in Figure 9. Very likely the problem is that they are not accounting for solid carbon as another product along with CO2 and CO – please see my previous comment.

5)      Alumina (the bare support) should also present some acid sites – the NH3-TPD for the support should be included in Figure 7.

6)      Why does Cr improve the dispersion of RuOx? This is an interesting result but unfortunately poorly explained in the manuscript.

7)      English must be substantially upgraded. See for example in the abstract “revealing that the activity of the 2Cr3Ru/Al2O3 catalyst had good stability” Activity and stability are not necessarily linked parameters. Use of English is poor in some sections of the manuscript

8)       Avoid the use of “conversion rates” there are no kinetic data in this study  - the term “conversion rate” is an unfortunate praxis in catalysis literature instead “conversion levels” is more adequate.

Author Response

(The authors gave the same response as above.)

Round 2

Reviewer 1 Report

The paper was properly revised according to referee's suggestions. Therefore, I can recommend this paper to be accepted for a publication on Catalysts.

Author Response

Thank you for your kind suggestion. 

Reviewer 2 Report

Reviewing of the revised version of the manuscript “Chromium-ruthenium oxides supported on gamma-alumina as an alternative catalyst for partial combustion of methane”

Comparing the revised and the first version of the manuscript, it is observable a considerable effort made by the authors for improving the manuscript. The new document include further details of the experimental section as well as paragraphs with important discussions.

However, an important concern arises from the way that authors still classifying the catalytic performance.

The comparison of the catalytic activity by means of the light-off curves (Figure 9- used to conclude that the 2Cr3Ru/Al2O3 catalysts “can be considered for commercial use”) is not the most suitable tool for establishing the conclusions proposed by the authors.

The comparison has to be carried out per active site and considering that different transition metals were used, not the same amount of atoms are present in 5 wt.% of Pt, 5 wt.% Ru or 5 wt.% Cr.

Of course the light-off curves are a first approach to study the catalysts but the TOF values have to be used for a real comparison.

If Pt or Ru atoms are the active sites, and the size of the Ru species is not strongly modified in the final catalysts, this has to be used to normalize the results of the light-off curves per active site. After that, it is expectable a different classification of the catalytic performance than that made by the authors.

Finally, a 24 h stability test still being too short to label the catalytic stability of the studied system as "excellent".  I do recommend do not use word. Just a description of this result would be suitable in this case.

As I said above, the deep revision of the manuscript is observable. However, for the publication, the cited concern has to be considered and considering the relevance of this, I suggest publication only after major revisions, since this implies modifications of the manuscript including probably figures or tables and new conclusions.

Author Response

Please see the file attached. 

Reviewer 3 Report

The authors have addressed most of the comments pointed out by the referees and the updated version of the paper has been remarkably improved 

Author Response

Thank you for your kind suggestion. 

Round 3

Reviewer 2 Report

After the revision of the new revised version, I found that the authors have addressed all the concerns generated in the previous versions. Therefore I do recommend the publication of the manuscript in the present form.